# Residual Group Channel and Space Attention Network for Hyperspectral Image Classification

**Peida Wu**, **Ziguan Cui** *, **Zongliang Gan and Feng Liu**

College of Telecommunications and Information Engineering, Nanjing University of Posts and Telecommunications, Nanjing 210003, China; 1218012329@njupt.edu.cn (P.W.); ganzongliang@gmail.com (Z.G.); liuf@njupt.edu.cn (F.L.)

\* Correspondence: cuizg@njupt.edu.cn

**Abstract:** Recently, deep learning methods based on three-dimensional (3-D) convolution have been widely used in the hyperspectral image (HSI) classification tasks and shown good classification performance. However, affected by the irregular distribution of various classes in HSI datasets, most previous 3-D convolutional neural network (CNN)-based models require more training samples to obtain better classification accuracies. In addition, as the network deepens, which leads to the spatial resolution of feature maps gradually decreasing, much useful information may be lost during the training process. Therefore, how to ensure efficient network training is key to the HSI classification tasks. To address the issue mentioned above, in this paper, we proposed a 3-DCNN-based residual group channel and space attention network (RGCSA) for HSI classification. Firstly, the proposed bottom-up top-down attention structure with the residual connection can improve network training efficiency by optimizing channel-wise and spatial-wise features throughout the whole training process. Secondly, the proposed residual group channel-wise attention module can reduce the possibility of losing useful information, and the novel spatial-wise attention module can extract context information to strengthen the spatial features. Furthermore, our proposed RGCSA network only needs few training samples to achieve higher classification accuracies than previous 3-D-CNN-based networks. The experimental results on three commonly used HSI datasets demonstrate the superiority of our proposed network based on the attention mechanism and the effectiveness of the proposed channel-wise and spatial-wise attention modules for HSI classification. The code and configurations are released at Github.com.

**Keywords:** hyperspectral image classification; convolutional neural network; attention mechanism; channel-wise attention; spatial-wise attention

---

## 1. Introduction

With the rapid development of remote sensing hyperspectral imaging technology, hyperspectral image has been studied and applied in more and more practical applications, including ocean research [1], vegetation analysis [2], road detection [3], geological disaster detection [4], and environmental analysis [5], etc. A hyperspectral image (HSI) contains abundant spectral and spatial information, which makes the HSI supervised classification task a hot research topic in the remote sensing analysis field. However, owing to the diversity of ground materials and the Hughes phenomenon coming from the increasing number of spectral bands [6], how to make full use of and extract the most discriminative features from spectral and spatial dimensions is a crucial issue in the HSI classification task.

In the past, traditional machine learning (ML)-based HSI classification methods mainly contain two steps, i.e., feature engineering and classifier training [7]. These methods usually focus on feature

selection and classifier design, which requires lots of manual design based on specific HSI data. For example, [8] divided bands into several sets by the cluster method and selected useful bands to construct tasks. Manifold ranking was introduced to eliminate the drawbacks of traditional salient band selection methods [9]. In [10], the Markov random field (MRF) was used in combination with band selection. However, inappropriate dimensionality reduction in the spectral domain from the manual design may lead to the loss of much useful spectral information. Therefore, while adopting SVM as the final classifier like [11], many papers also suggested to explore input data with more spatial information in the feature engineering to improve the classification performance [12,13]. For instance, [14] developed a region kernel to measure the region-to-region distance similarity and extract spectral-spatial combined features. A common problem among these methods is that traditional ML-based methods usually cannot make full use of ground material feature expression due to the difficulty in designing feature extraction methods. Therefore, traditional ML-based methods usually cannot achieve a high classification performance.

In recent years, deep learning (DL) has shown a powerful ability to extract hierarchical and nonlinear features, and DL methods based on the convolutional neural network (CNN) have been widely used in HSI classification tasks. So far, many works based on CNN have demonstrated that the end-to-end approach can reduce the possibility of the information loss during data preprocessing and improve the classification accuracy by learning deep features. For example, a unified framework combining CNN with a stacked autoencoder (SAE) was proposed to adaptively learn weight features for each pixel by one-dimensional (1-D) convolutional layers [15,16]. In [17], SAE was also used to capture the representative stacked features. However, the input data of these 1-D-CNN-based methods must be flattened into 1-D vectors, which means that they cannot make full use of the spatial contextual relationship between pixels from raw HSI data.

To solve the above problems, the two-dimensional convolution neural network (2-D-CNN) was introduced to extract spectral and spatial features at the same time in many papers. For example, [18] proposed a multiscale covariance maps (MCMs)-based feature extraction method, and combined it with the 2-D-CNN model to integrate the spectral and spatial information. However, the proposed method required specific hand-crafted feature extraction for different HSI datasets, which was difficult to design precise and complete artificial features. Then, a contextual deep CNN was introduced in [19] to explore local contextual interactions. It used 2-D-CNN to extract spectral and spatial information separately. However, in these methods, when the network is deep, the rapid increase of network parameters will cause these 2-D-CNN-based models to be difficult to train. It may cause a degradation problem and finally lead to low classification accuracies. Therefore, residual connections were used to alleviate this phenomenon. The authors of [20] adopted residual learning to optimize several convolutional layers as the identity mapping, and constructed a very deep network to extract spectral and spatial features. Since HSIs have both spectral and spatial information, [21] proposed an end-to-end spectral-spatial residual network (SSRN), which consists of spectral and spatial residual blocks consecutively, to learn spectral and spatial features, respectively. In addition, to obtain a lower training cost and parameter scale, inspired by the densely connected convolutional network [22], an end-to-end spectral-spatial dual-channel dense network (SSDC-DenseNet) was proposed to reduce the model scale and explore high-level features [23]. Due to the densely connected structure, each layer will accept feature maps from all previous layers as its additional input data. Though these HSI classification methods based on 2-D-CNN could utilize the spatial context information, they separated spectral-spatial joint features into two independent learning parts. Since HSI is a 3-D data cube, it means that these methods neglect the close correlations between spectral information and spatial information.

Therefore, some three-dimensional convolution neural networks (3-D CNNs) models were proposed to learn spectral-spatial joint features directly from raw HSI data. With the help of residual connections, [24] constructed a three-dimensional residual network (3-D-ResNet) to improve the classification performance. The experimental results demonstrated that 3-D-ResNet could mitigate the declining accuracy effect and achieved promising classification performance with few training

samples. The authors of [25] further studied 3-D CNNs to extract spectral-spatial combined features by using input cubes of HSIs with a smaller spatial size. Based on 3-D CNN and the densely connected convolutional network [23], [26] proposed the three-dimensional densely connected convolutional network (3-D-DenseNet) for HSI classification. The network could become very deep and extract more representative spectral-spatial combined features. To further reduce the training time, [27] proposed an end-to-end fast dense spectral-spatial convolution (FDSSC) by using a dynamic learning rate and parametric rectified linear units. To reduce the number of the parameters and solve the imbalance of classes, [28] used 3-D-ResNeXt and the label smoothing strategy to simultaneously extract spectral and spatial features, and achieved obvious classification performance improvement. To extract the spectral-spatial features of different scales, [29] designed a multiscale octave 3-D CNN with channel and spatial attention (CSA-MSO3-DCNN). 3-D-CNN convolution kernels of different sizes could capture diverse features of HSI data. These 3-D-CNN-based methods can indeed make full use of the original characteristics of raw HSI data and the correlation between spectral and spatial information. Furthermore, the graph convolutional network [30] was applied to alleviate the deficient labeled samples in [31]. The spatial information was added into the approximate convolutional operation on the graph signal. So, the features obtained by the graph convolutional network made full use of both the adjacency nodes in the graph and neighbor pixels in the spatial domain. However, the features processed by the convolutional layers may contain much useless or disturbing information. If these useless features are sent directly to the next layer without any process, as the network is going deeper, the learning efficiency of the network will be lower, and will finally affect the classification performance. Therefore, how to deal with the feature maps after convolutional layers and pay more attention on those features with a large amount of useful information is another key for HSI classification tasks.

Recently, many classical and effective computer vision methods have been embedded in CNN to improve the performance of DL models. Among them, the CNN model fused with the attention mechanism delivers promising outcomes in improving HSI classification performance. The goal of the attention mechanism is to focus on salient features or regions with a large amount of information. Through a series of weight coefficients, the CNN model with the attention mechanism could improve the quality of feature maps after convolutional layers. For example, to extract more discriminative spectral and spatial features, [32] combined FDSSC [27] and the convolutional block attention module (CBAM) [33], and proposed a double-branch multi-attention mechanism network (DBMA) for HSI classification, which consists of two parallel branches using channel-wise and spatial-wise attention separately. The experimental results demonstrated the effectiveness of channel-wise and spatial-wise attention. However, the parallel branching method did not take the correlation between spectral and spatial information into consideration, so DBMA did not obtain a satisfactory classification accuracy. In order to introduce global spatial information and solve the locality of the convolution operation, the self-attention mechanism [34] was introduced to construct the non-local neural network. In [35], this attention module was attached to the spectral-spatial attention network (SSAN). This attention module, which only focuses on spatial information, cannot globally optimize feature maps processed by convolutional layers. The authors of [36] adopted the squeeze-and-excitation network (SENet) [37] to adaptively recalibrate channel feature responses by explicitly modelling interdependencies between channels. In addition, [38] also constructed a spatial-spectral squeeze-and-excitation (SSSE) module based on SENet. There was a problem that the channel features processed by the SSSE module contain much redundant information, which may affect the classification performance. To generate high-quality samples containing a complex spatial-spectral distribution, [39] proposed a symmetric convolutional GAN based on collaborative learning and the attention mechanism (CA-GAN). The joint spatial-spectral attention module could emphasize more discriminative features and suppress less useful ones. However, although these methods based on the attention mechanism can play a role in optimizing features, they all have limitations: For one thing, the network can only be optimized in a specific spectral or spatial dimension; for another thing, as the network deepens, the channel attention module may lose much useful information because the optimization is operated on the whole channels.

Our motivation was to construct a 3-DCNN-based network using an efficient attention module to solve the problem mentioned above. Inspired by the principle of SENet [37] and the bottom-up top-down structure that has been applied to image segmentation [40], we proposed a 3-DCNN-based residual group channel and space attention network (RGCSA) for HSI classification (The code and configurations are released at https://github.com/Lemon362/RGCSA-master). The framework consists of several building blocks with the same topology. Each block contains a convolutional layer for learning features, a residual group channel-wise attention module, and a residual spatial-wise attention module. According to the principle of bottom-up top-down, we unified the channel attention module and space attention module into the same structure, but their implementation methods of up-sampling are different. To optimize features in the channel dimension to the greatest extent and reduce the loss of useful information, we introduced the principle of grouping into the channel-wise attention module to realize the group channel attention mechanism. Compared with 3-D-ResNeXt [28], we only need fewer training samples to achieve a better classification accuracy by using the attention mechanism. When compared with DBMA [32] and SSAN [35], our network has fully optimized the feature maps processed by each convolutional layer in the channel dimension and spatial dimension, and shows an effective improvement for HSI classification.

In short, the three major contributions of this paper are listed as follows:

1. Combining the bottom-up top-down attention structure with the residual connection, we constructed residual channel and space attention modules without any additional manual design, and proposed a 3-DCNN-based residual group channel and space attention network (RGCSA) for HSI classification. On the one hand, residual connection can accelerate the flow of information, making the network better training. On the other hand, the structure of channel-wise attention first and then spatial-wise attention could strengthen important information and weaken unimportant information during the training process, and compared to the previous methods, RGCSA can achieve a higher HSI classification accuracy with fewer training samples.

2. We applied the principle of group convolution to the channel attention structure to construct a residual group channel attention module, which aims to emphasize each piece of useful channel information. Compared with the previous channel attention methods, it can reduce the possibility of losing useful channel information during attention optimization.

3. We proposed a novel spatial-wise attention module, which utilized transposed convolution as an up-sampling method. It ensures the mapping relationship of spatial pixels in the attention optimization process, and makes full use of context information to optimize the features in the spatial dimension to focus on the most informative areas.

The remaining sections of this paper are organized as follows. Section 2 illustrates the related work about our proposed network for HSI classifications. Section 3 presents a detailed network configuration of the overall framework and individual modules. Then, Section 4 illustrates experimental datasets and the parameter setting, and then shows the experimental results and analyses. Finally, in Section 5, we summarize some conclusions and introduce future work.

## 2. Related Work

In this section, we first introduce the proposed end-to-end pixel-level HSI classification framework and some basic knowledge, including ResNeXt and SeNet, and then introduce the architecture of our proposed residual group channel and space attention mechanism in detail.

### 2.1. Pixel-Level HSI Classification Framework

The proposed end-to-end pixel-level HSI classification architecture explores spectral-spatial combined information by 3-D-CNN, which can make full use of the close correlation between these two dimensions. To fully utilize the whole spatial information, we adopted zero padding in the four directions of the spatial dimension of the raw HSI data. Therefore, the raw HSI data cube is converted

into $X \in \mathbb{R}^{w \times w \times b}$, where $w$ represents the spatial width and height, and $b$ represents the number of spectral bands. In this framework, all available labeled data are divided into three groups: Training dataset, validation dataset, and testing dataset for each HSI dataset. Firstly, the network is fed into a small batch of data to train the model for hundreds of epochs. During the training process, the label smoothing strategy is used in the cross-entropy loss function to alleviate the problem of the imbalance of classes. Then, at the same time, the validation dataset monitors the whole training process by computing the classification accuracy every few epochs; in this way, the network can choose the best model with the highest accuracy. Finally, the testing dataset is adopted to evaluate the classification performance of proposed network.

### 2.2. Three-Dimensional ResNeXt Network

In the HSI classification task, in order to mitigate the decreasing-accuracy phenomenon and reduce the huge number of parameters caused by 3-D-CNN, 3-D-ResNeXt was first proposed in [28] and achieved high classification accuracy.

As shown in Figure 1, with the growing number of hyperparameters (width, filter sizes, strides, etc.), 3-D-CNN will lead to a dramatic increase in computational cost, especially when there are multiple layers. Therefore, with the split-transform-merge strategy, [41] split the CNN layer of ResNet into several groups, so that each feature learning process was performed in a low-dimensional embedding, and each output feature map was aggregated by summation performing in a high-dimensional embedding. From Figure 1, we can find that the ResNeXt with *cardinality* = 8 has roughly the same complexity as the ResNet. This operation makes it possible to reduce the number of parameters to a large extent when building a deep network through 3-D convolutional layers. Therefore, 3-D-ResNeXt is a suitable choice in the HSI classification networks based on 3-D-CNN.

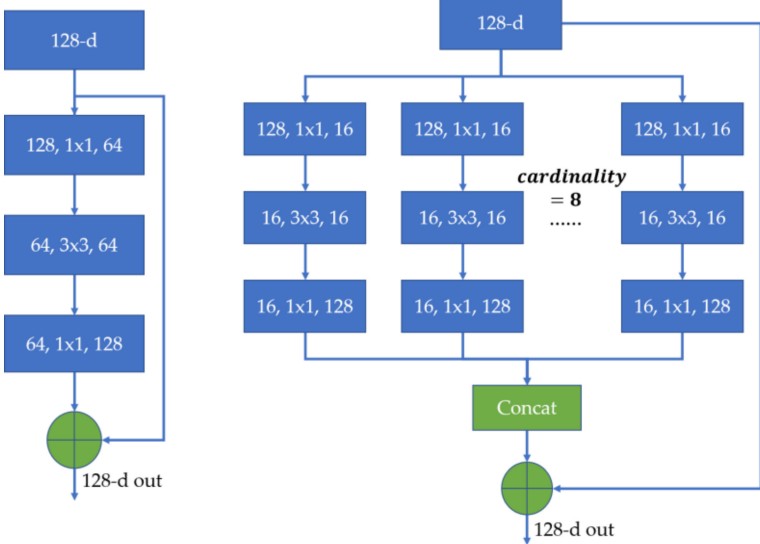

**Figure 1.** A block of ResNet (Left) and ResNeXt with cardinality = 8 (Right). A layer is shown as (# in channels, filter size, # out channels).

However, there is a problem with the 3-D-ResNeXt network proposed in [28] that 3-D-ResNeXt could not optimize the output feature maps of each convolutional layer during the training process. It may cause lots of useless information to be sent to the next layer, which seriously affects the efficiency of network training. Therefore, in this paper, on the basis of using 3-D-ResNeXt to extract spectral-spatial features, we will pay more attention on how to optimize the feature maps extracted by 3-D-ResNeXt in the channel dimension and spatial dimension by the attention mechanism.

### 2.3. Squeeze-and-Excitation Network

As we all know, the convolution operation is the core of convolutional neural networks (CNNs), which enables networks to extract informative features from different dimensions. Now, many researchers focus on how to strengthen the representational power of a CNN, and [37] proposed a novel architectural unit focusing on the channel relationship, named the squeeze-and-excitation (SE) block.

The structure of the SE block is shown in Figure 2. The SE block consists of a global pooling layer, two fully connected (FC) layers, and two activation function layers (one is ReLU, and the other is Sigmoid). The principle of the SE block is to enhance the important features and weaken the unimportant features by controlling the weight coefficient of each channel. First, the global average pooling (GAP) layer implements the squeeze process. To take advantage of the correlation of channels, GAP averages the spatial dimension of feature maps with a size of $H \times W \times C$ ($H$ and $W$ represent the two dimensions in space, and $C$ represents the number of channels) to form $1 \times 1 \times C$ feature maps, which can shield the spatial distribution information and integrate global spatial information to obtain the importance of the feature channel. Then, the excitation process contains two FC layers. The first FC layer is used to compress $C$ channels into $C/r$ channels and the second one restores the compressed feature map to the original size of $1 \times 1 \times C$. Finally, by multiplying the weight coefficients limited by Sigmoid to the [0, 1] range with the original output feature maps, it can be ensured that the input features of the next layer are optimal in the channel-wise dimension.

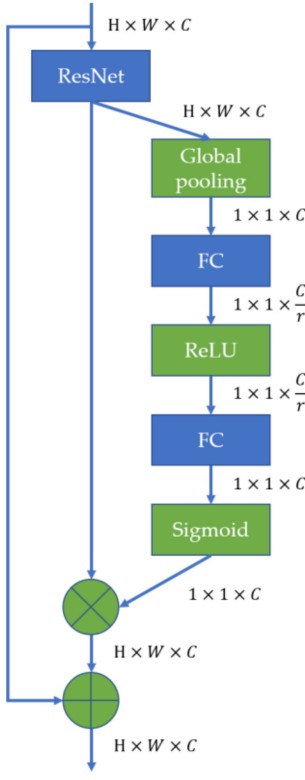

**Figure 2.** The structure of the SE block inserted into the ResNet.

### 2.4. Proposed Attention Mechanism

In general, the purpose of the attention mechanism is to strengthen important information and weaken unimportant information by certain methods. Inspired by the principle of SENet [36], we used the bottom-up top-down structure to construct the attention mechanism based on CNN. Our proposed channel-wise and spatial-wise attention modules have the same architecture, but the implementation methods of up-sampling are different. In addition, due to the superposition of attention modules, the network becomes deeper and some feature maps may be weakened, which may destroy the original

features of input and result in a declining accuracy effect. Therefore, to alleviate this problem, we also introduced the residual connection to the attention module to form the residual attention module.

The basic structure of the proposed residual attention module is shown in Figure 3. The whole residual attention module consists of two parts: One is the attention module and the other is the residual connection part. In the attention module, the down-sampling layer compresses the features extracted by 3-D-CNN in the corresponding dimension to learn the compact features. Then, these feature maps are sent to the up-sampling layer to restore the original size by some methods. It ensures that the subsequent calibration feature process can accurately assign the weight coefficient to the corresponding position. Finally, the Sigmoid function is used to limit the optimized feature attention maps to [0, 1] to obtain the weight coefficient corresponding to each position. The coefficients tending to 0 indicate that the amount of information at this position is small, while those tending to 1 show that this position has more important information. Therefore, we only need to multiply the output with the feature maps from 3-D-CNN to rescale the final output of the attention module. In this way, these weights are assigned to each feature map to achieve adaptively recalibrating features. With the superposition of attention modules, more important features can be strengthened and become clearer while unimportant features will be gradually weakened to 0 so as not to affect the network training process. Furthermore, to allow this attention module to be inserted into deep networks, we add the residual connection and an extra convolutional layer to prevent the original features from being destroyed and accelerate the flow of information.

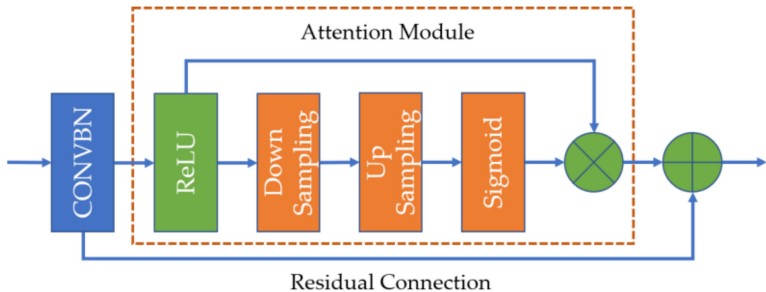

**Figure 3.** The basic structure of the proposed residual attention module.

In our proposed RGCSA, the network has two attention modules based on the above residual attention structure: Residual group channel-wise attention module (RGCA) and residual spatial-wise attention (RSA) module. These two attention modules are all based on CNN operation and the bottom-up top-down attention structure. The specific implementation details of the attention module will be presented in the following paragraphs.

### 2.4.1. Residual Group Channel-Wise Attention Module

The channel-wise attention module mainly refines the channel weights of the feature maps. Since each channel of the feature maps is considered as a feature detector, channel attention focuses on the meaningful channels and decreases the meaningless channels. Figure 4 shows the structure of the proposed residual group channel-wise attention (RGCA), which consists of $G$ residual channel-wise attention (RCA) blocks, where $G$ is the number of groups. Therefore, we will give a detailed introduction to the structure of the RCA block.

In our proposed residual channel-wise attention (RCA) module, which is shown in Figure 5, we first use global average pooling (GAP) and reshape the operation to shield the information of the spectral and spatial dimensions to obtain feature maps with a size of $1 \times 1 \times 1$, $C$, where $C$ represents the number of channels. Then, the feature tensors are sent to a $1 \times 1 \times 1$, $C/r$ 3-D convolutional layer instead of fully connected layer to reduce the channel dimensionality and extract abstract features with more important information. After convolution operation, the number of channels becomes $C/r$, where $r$ is a reduction ratio. Then, a ReLU layer is used to strengthen the nonlinear relationship of

channel responses. Next is the up-sampling process. Here, we still use a $1 \times 1 \times 1$, $C$ 3-D convolutional layer to increase the channel dimension and finally generate $C$ feature maps. As mentioned above, a Sigmoid function is applied to limit the range of features, and the output is multiplied with the feature maps from 3-D-CNN to obtain the channel-refined features.

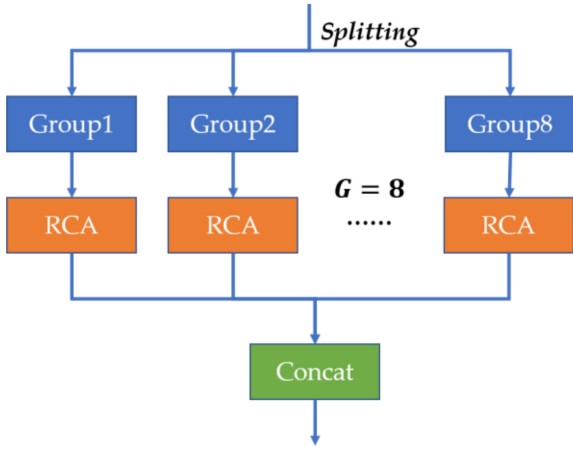

**Figure 4.** The structure of the residual group channel-wise attention (RGCA) module.

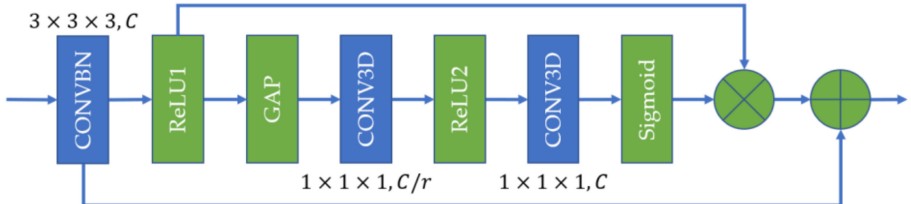

**Figure 5.** The structure of the residual channel-wise attention (RCA) module.

In addition, if we directly optimize the whole channel, as the network deepens, the weight coefficients of some originally useful channel features may decrease, so that this part of important information is lost. Therefore, we introduce the principle of group convolution of 3-D-ResNeXt to the RCA module to construct the final residual group channel-wise attention (RGCA) module. From Figure 4, we can see that the feature maps are divided into $G$ groups, and then sent to each RCA module respectively. In this way, we can ensure that each channel of the feature maps in the deep network is rescaled to the optimal value, and reduce the possibility that useful features may be lost during the optimization process.

The residual group channel-wise attention module is added after the 3-D-CNN layer of 3-D-ResNeXt but before the summation operation for residual connection, which is shown on the left side of Figure 6. In this paper, to simplify the complexity of the network module design, we set the group number $G$ of RGCA to the same value as the parameter *cardinality* of 3-D-ResNeXt. In this way, we can conveniently and effectively combine the grouping operation of RGCA with 3-D-ResNeXt to form the structure on the right side of Figure 6, which greatly simplifies the complexity of the network.

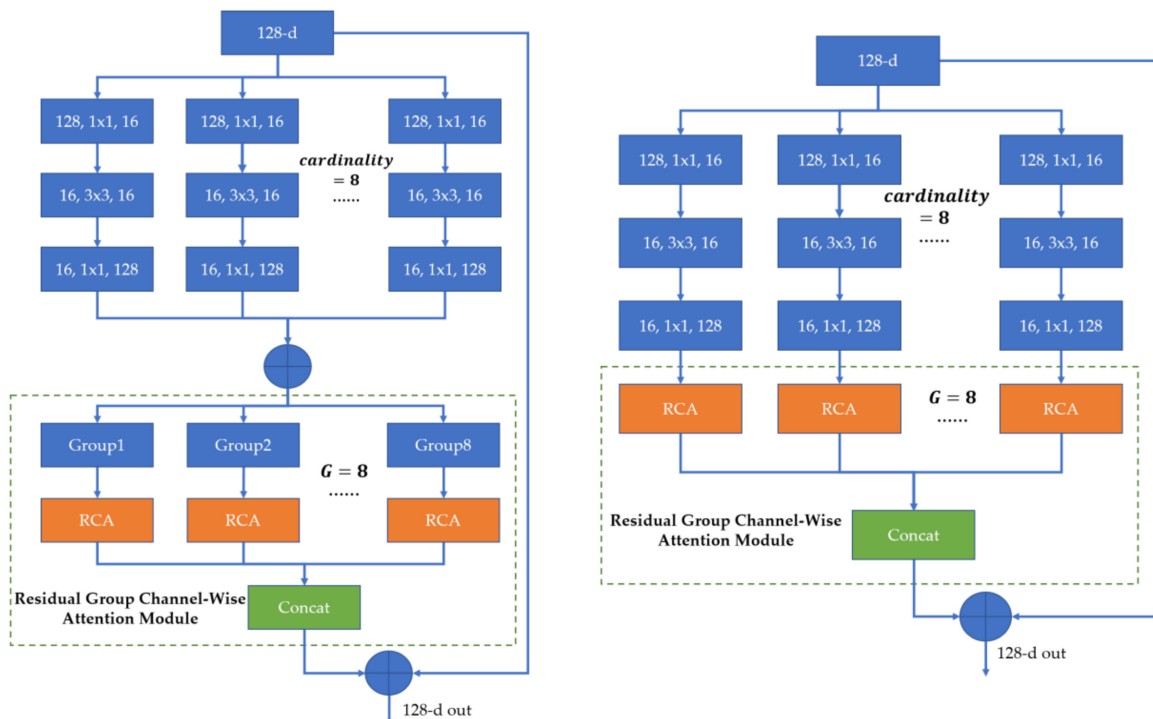

**Figure 6.** The structure of the residual group channel-wise attention (RGCA) module before simplification (**left**) and after simplification (**right**).

### 2.4.2. Residual Spatial-Wise Attention Module

Compared with the channel-wise attention module, the spatial-wise attention pays attention to the informative region of the spatial dimension. To optimize the spatial features, we proposed a novel spatial attention module, and the structure of the proposed residual spatial-wise attention (RSA) module is shown in Figure 7.

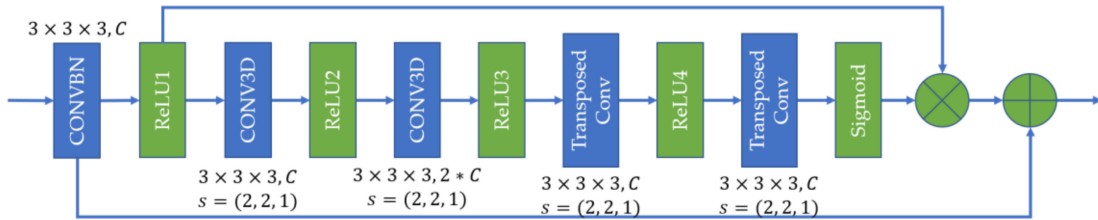

**Figure 7.** The structure of the residual spatial-wise attention (RSA) module.

In the residual spatial-wise attention (RSA) module, we also use 3-D-CNN with stride = (2, 2, 1) as the down-sampling layer to reduce the spatial dimension of features while keeping the spectral dimension unchanged. After two down-sampling layers, we obtain the feature maps containing important spatial information. Then, we introduce a novel way (transposed convolution) to restore the original size. Transposed convolution is a special kind of forward convolution: First, the size of the input feature map is expanded by zero-filling operation according to a certain ratio; then, the convolution kernel is rotated by 180°, which is equivalent to transposing the convolution kernel matrix; finally, the normal convolution operation is performed by this new convolution kernel. Transposed convolution can maintain the mapping relationship of spatial positions before and after operation when restoring the size, which is important for the subsequent weight optimization process.

Inspired by the structure of CBAM [33], the residual spatial-wise attention module is added after the residual group channel-wise attention module to form an optimized structure of the channel first

and space second. Additionally, the experiment proves that this sequential attention mechanism can obtain a higher classification accuracy with fewer training samples than 3-D-ResNeXt.

## 3. Network Configuration and Experimental Setup

In this section, taking the Indian Pines (IN) dataset as an example, we give an introduction of the overall framework and present the network configuration of each module and experiment setup in detail.

### 3.1. Overall Framework of the Proposed Network

Figure 8 shows the network structure of the proposed residual group channel and space attention (RGCSA) network. In Figure 8, we take the Indian Pines (IN) dataset with $16 \times 16$ patch as the input to illustrate the size of the feature maps used in our network. It consists of three major modules: The first is the initialization module, which is used to initially reduce the spectrum dimension by a 3-D convolutional layer; the second is the residual group channel and space attention module; and the third is the classification module.

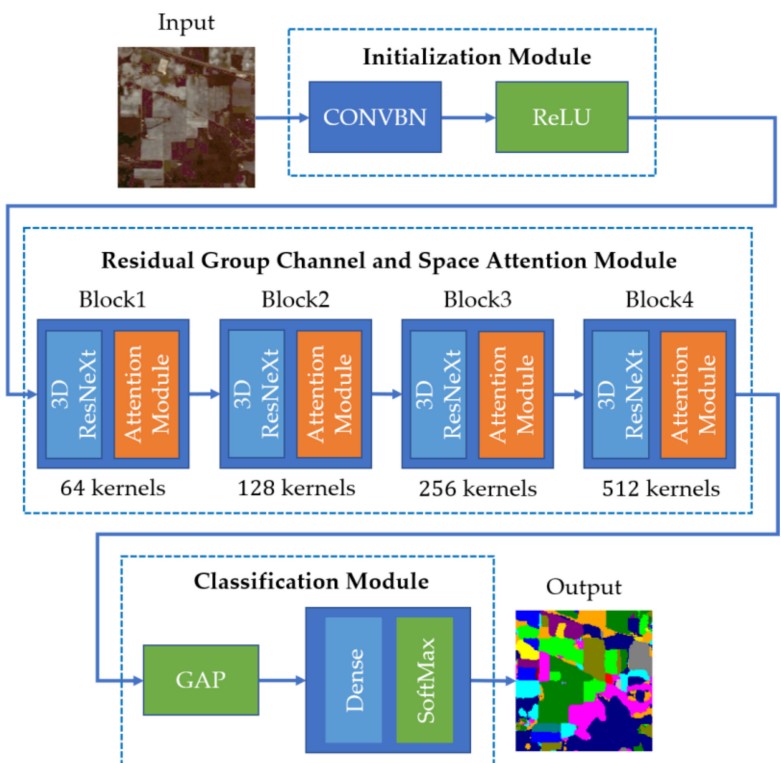

**Figure 8.** Overall HSI classification structure of the proposed residual group channel and space attention (RGCSA) network.

The residual group channel and space attention module contains four of the same building blocks, whose filters numbers are {64, 128, 256, 512}, respectively. We use 3-D-ResNeXt to extract the spectral-spatial features, and insert the proposed RGCSA into the 3-D-ResNeXt to optimize the features. In addition, in block 4, we did not use the residual spatial-wise attention module because the spatial dimension in block 4 is too small to perform a good dimensionality reduction operation. In short, the features of the first three modules are optimized by RGCSA and the last module only contains RGCA. In the classification module, we use the global average pooling (GAP) layer to replace the fully connected layer (FC) to greatly reduce the number of parameters and improve the network training efficiency. Furthermore, we also introduced the label smoothing strategy proposed in [28] to solve the imbalance of classes.

Next, we will introduce the specific network configuration of each module.

### 3.2. Network Configuration of the Proposed Network

First, we introduce the network configuration of the overall RGCSA, and the network configurations and parameter settings of the RGCA and RCA module are introduced later. Taking the IN dataset as an example, the detailed network parameter setting of the proposed RGCSA network for three HSI datasets is shown in Table 1.

**Table 1.** The network configuration of the proposed RGCSA network.

| Layer | Output Size | RGCSA | Connected to |
|---|---|---|---|
| Input | $16 \times 16 \times 200$ | | |
| CONVBN | $16 \times 16 \times 100, \ 32$ | $3 \times 3 \times 7, \ 32 \ conv$<br>$s = (1, \ 1, \ 2)$ | Input |
| Block1 | $16 \times 16 \times 100, \ 64$ | $3 \times 3 \times 3, \ 64 \ conv$<br>same | CONVBN |
| Block2 | $8 \times 8 \times 50, \ 128$ | $3 \times 3 \times 3, \ 128 \ conv$<br>$s = (2, \ 2, \ 2)$ | Block1 |
| Block3 | $4 \times 4 \times 25, \ 256$ | $3 \times 3 \times 3, \ 256 \ conv$<br>$s = (2, \ 2, \ 2)$ | Block2 |
| Block4 | $2 \times 2 \times 13, \ 512$ | $3 \times 3 \times 3, \ 512 \ conv$<br>$s = (2, \ 2, \ 2)$ | Block3 |
| GAP | 512 | | Block4 |
| Dense (SoftMax) | 16 | 16 | GAP |

Here, taking block 1 as an example, Figure 9 shows the structure of the building block. First, the $1 \times 1 \times 1$ convolutional layer changes the dimensionality of the feature channel. Then, the input tensor is transformed to $G$ groups through the splitting operation. These feature maps of each group are sent to the 3-D-CNN layer with $3 \times 3 \times 3$ convolutional kernel to extract spectral-spatial features. Then, the RGCA is used to focus on the important channel features of each group, which are aggregated into a high-dimensional feature vector again through a concatenate layer. The feature maps optimized by RGCA are fed into RSA to pay attention to the spatial dimension to strengthen the important information region and weaken unimportant region information. Finally, the residual connection allows the original input features to be fused with the processed features at the same size.

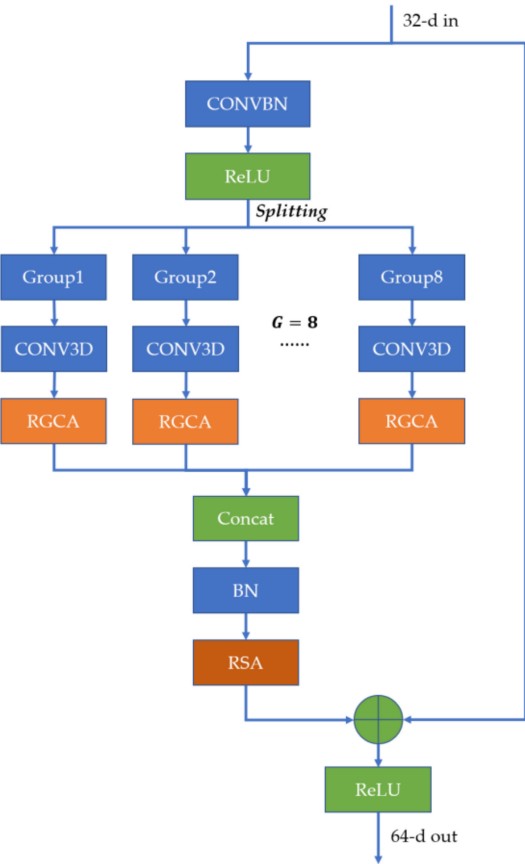

**Figure 9.** General structure of the building block in RGCSA (taking block 1 as an example).

### 3.2.1. Network Configuration of the Residual Group Channel-Wise Attention Module

Since RGCA consists of $G$ group RCA, according to the structure of RCA shown in Figure 5, we only select one group to introduce the network configuration of the residual channel-wise attention module which is shown in Table 2. First of all, followed by the 3-D-CNN layer of 3-D-ResNeXt, a convolutional layer with a size of $3 \times 3 \times 3$, 8 and the 'same' padding method is used to further extract abstract features. After the first ReLU activation layer, feature maps with the shape of $16 \times 16 \times 100$, 8 are obtained. Then, the global average pooling layer and reshape operation are used to flatten the spectral and spatial dimensions to obtain $1 \times 1 \times 1$, 8 feature maps. Setting the reduction ratio $r = 4$, $1 \times 1 \times 1$, 2 3-D-CNN is used to reduce the number of channels. To perform the up-sampling operation, feature maps are processed by $1 \times 1 \times 1$, 8 3-D-CNN again. Finally, the optimized feature vectors ranging from 0 to 1 are multiplied by the features processed by $3 \times 3 \times 3$, 8 3-D-CNN, and the residual connection is used to add the original input from 3-D-ResNeXt and these optimized features. After the channel attention module, the important channel is highlighted while the unimportant channel is suppressed.

**Table 2.** The network configuration of the residual group channel-wise attention (RGCA) module.

| Layer | Output Size | RGCA | Connected to |
|---|---|---|---|
| CONV3D | $16 \times 16 \times 100,\ 8$ | | Group1 |
| CONVBN | $16 \times 16 \times 100,\ 8$ | $3 \times 3 \times 3,\ 8\ conv$ same | CONV3D |
| ReLU1 | $16 \times 16 \times 100,\ 8$ | | CONVBN |
| GAP (Reshape) | $1 \times 1 \times 1,\ 8$ | | ReLU1 |
| CONV3D | $1 \times 1 \times 1,\ 2$ | $1 \times 1 \times 1,\ 2$ | GAP |
| ReLU2 | $1 \times 1 \times 1,\ 2$ | | CONV3D |
| CONV3D | $1 \times 1 \times 1,\ 8$ | $1 \times 1 \times 1,\ 8$ | ReLU2 |
| Sigmoid | $1 \times 1 \times 1,\ 8$ | | CONV3D |
| Multiply | $16 \times 16 \times 100,\ 8$ | | Sigmoid, ReLU1 |
| Add | $16 \times 16 \times 100,\ 8$ | | Multiply, CONV3D |

### 3.2.2. Network Configuration of the Residual Spatial-Wise Attention Module

The network configuration of the residual spatial-wise attention module in block 1 is described in Table 3. First, like the residual channel-wise attention module, we use $3 \times 3 \times 1,\ 64$ 3D-CNN with the 'same' padding method to learn spatial features while keeping the spectral dimension unchanged. Then, two convolutional layers with a size of $3 \times 3 \times 1$, *stride* $= (2,\ 2,\ 1)$, and {64, 128} filters, respectively, are used to focus on the important spatial information and reduce the spatial dimension. Then, two $3 \times 3 \times 1,\ 64$ transposed convolutional layers with *stride* $= (2,\ 2,\ 1)$ realize the up-sampling operation. Finally, the layer implement optimization and fusion operations are multiplied and added, respectively. In addition, in block 3 of the RGCSA network, since the feature dimension becomes $4 \times 4 \times 25$, we set the size of 3-D-CNN, which is used to focus on the important spatial information, to $1 \times 1 \times 1$. Additionally, in block 4, we do not add this residual spatial-wise attention module to optimize the spatial information.

**Table 3.** The network configuration of the residual spatial-wise attention (RSA) module.

| Layer | Output Size | RSA | Connected to |
|---|---|---|---|
| BN | $16 \times 16 \times 100,\ 64$ | | Concat |
| CONVBN | $16 \times 16 \times 100,\ 64$ | $3 \times 3 \times 1,\ 64\ conv$ same | BN |
| ReLU1 | $16 \times 16 \times 100,\ 64$ | | CONVBN |
| CONV3D | $8 \times 8 \times 100,\ 64$ | $3 \times 3 \times 1,\ 64\ conv$ $s = (2,\ 2,\ 1)$ | ReLU1 |
| ReLU2 | $8 \times 8 \times 100,\ 64$ | | CONV3D |
| CONV3D | $4 \times 4 \times 100,\ 128$ | $3 \times 3 \times 1,\ 128\ conv$ $s = (2,\ 2,\ 1)$ | ReLU2 |
| ReLU3 | $4 \times 4 \times 100,\ 128$ | | CONV3D |
| Transposed Conv | $8 \times 8 \times 100,\ 64$ | $3 \times 3 \times 1,\ 64\ conv$ $s = (2,\ 2,\ 1)$ | ReLU3 |
| ReLU4 | $8 \times 8 \times 100,\ 64$ | | Transposed Conv |
| Transposed Conv | $16 \times 16 \times 100,\ 64$ | $3 \times 3 \times 1,\ 64\ conv$ $s = (2,\ 2,\ 1)$ | ReLU4 |
| Sigmoid | $16 \times 16 \times 100,\ 64$ | | Transposed Conv |
| Multiply | $16 \times 16 \times 100,\ 64$ | | Sigmoid, ReLU1 |
| Add | $16 \times 16 \times 100,\ 64$ | | Multiply, BN |

### 3.3. Experimental Setup

We tested the factors that affect the HSI classification performance of the proposed network (i.e., the ratios of the training, validation, and test datasets for different HSI datasets, and the number of groups *G*), and the experimental results compared with several widely used methods are illustrated in Section 4. Finally, the most suitable ratios were 3 : 1 : 6 for the Indian Pines (IN) dataset, and 2 : 1 : 7

for the Pavia University (UP) and Kennedy Space Center (KSC) datasets. The number of groups $G$ in RGCA was 8, and the reduction ratio $r$ of RCA was 4. RMSProp was adopted as the optimizer to minimize the cross-entropy loss function. The initial learning rate was set to 0.0003. All the training and testing results were obtained on the same computer, with the configuration of 32GB of memory, NVIDIA GeForce GTX 1070 8GB, and Intel i7 7820HK.

## 4. Experiments and Results

In this section, we first introduce three HSI datasets used in this paper, i.e., the Indian Pines (IN) dataset, the Pavia University (UP) dataset, and the Kennedy Space Center (KSC) dataset. Then, we discuss the two main factors affecting the classification performance. Finally, we compare the proposed RGCSA network with several representative HSI classification models, which are introduced in Section 1, i.e., SVM [11], SSRN [21], 3-D-ResNeXt [28], DBMA [32], and SSAN [35]. The overall accuracy (OA), average accuracy (AA), and kappa coefficient (Kappa) are used as the indicators to measure HSI classification performance. OA refers to the ratio of the number of correct classifications to the total number of HSI pixels in the test datasets. AA refers to the average accuracy of all classes. The kappa coefficient is an indicator used for the consistency test between the classification results and ground truth, and it can also be used to measure the classification accuracy. In short, the higher values of these three indicators represent the better classification results. Let $M \in \mathbb{R}^{N \times N}$ represent the confusion matrix of classification results, where $N$ is the number of land-cover categories. According to [35], the values of OA, AA, and kappa can be calculated as follows:

$$OA = sum(diag(M))/sum(M), \tag{1}$$

$$AA = mean(diag(M))./sum(M,2), \tag{2}$$

$$Kappa = \frac{OA - (sum(M,1) \times sum(M,2))/(sum(M))^2}{1 - (sum(M,1) \times sum(M,2))/(sum(M))^2}, \tag{3}$$

where $diag(M) \in \mathbb{R}^{N \times 1}$ is a vector of the diagonal elements of $M$, $sum(\cdot) \in \mathbb{R}^1$ represents the sum of all elements of the matrix, $sum(\cdot, 1) \in \mathbb{R}^{1 \times N}$ represents the sum of elements in each column $sum(\cdot, 2) \in \mathbb{R}^{N \times 1}$ represents the sum of elements in each row, $mean(\cdot) \in \mathbb{R}^1$ represents the mean of all elements, and ./ represents the elementwise division.

To obtain a statistical evaluation, we repeated each experiment 5 times, and calculated the mean value as the final result.

### 4.1. Experimental Datasets

We used three available commonly used HSI datasets [42] in our experiment to evaluate the classification performance of the proposed RGCSA model.

The Indian Pines (IN) dataset [20] was collected by the Airborne Visible/Infrared Imaging Spectrometer (AVIRIS) in 1992 from Northwest Indiana. It contains 16 classes with the size of $145 \times 145$ pixels and a spatial resolution of 20 m by pixel. There are 220 bands in the wavelength range of 0.4 to 2.5 um. Since 20 bands are corrupted by water absorption effects; the remaining 200 bands can be adopted for HSI experiments.

The Pavia University (UP) dataset [20], gathered by Reflective Optics System Imaging Spectrometer (ROSIS) in 2001 in the Pavia region of northern Italy, has $610 \times 340$ pixels with a resolution of 1.3 m by pixel, and contains 9 vegetation classes. Since 12 bands with strong noise and water vapor absorption were removed, 103 bands ranging from 0.43 to 0.86 um were adopted for analysis.

The Kennedy Space Center (KSC) [43] was firstly acquired by AVIRIS in 1996 in the Kennedy Space Center, containing 224 bands with center wavelengths in the range of 0.4 to 2.5 um. The image has $512 \times 614$ pixels with a spatial resolution of 18 m and 13 types of geographic objects. Since

water absorption and low signal-to-noise ratio (SNR) bands were removed, 176 bands were adopted for analysis.

Tables 4–6 list the total number of samples of each class in each dataset and the number of training, validation, and test samples of three datasets under the optimal ratios, which obtained the best classification performance, i.e., 3 : 1 : 6 for the IN dataset, and 2 : 1 : 7 for the UP and KSC datasets.

**Table 4.** The number of training, validation, test, and total samples in the IN dataset.

| No. | Class | Train | Val | Test | Total Samples |
|-----|-------|-------|-----|------|---------------|
| 1 | Alfalfa | 14 | 1 | 31 | 46 |
| 2 | Corn-notill | 429 | 131 | 868 | 1428 |
| 3 | Corn-mintill | 249 | 83 | 498 | 830 |
| 4 | Corn | 72 | 22 | 143 | 237 |
| 5 | Grass-pasture | 145 | 42 | 296 | 483 |
| 6 | Grass-trees | 220 | 69 | 441 | 730 |
| 7 | Grass-pasture-mowed | 9 | 3 | 16 | 28 |
| 8 | Hay-windrowed | 144 | 55 | 279 | 478 |
| 9 | Oats | 6 | 4 | 10 | 20 |
| 10 | Soybean-notill | 292 | 94 | 586 | 972 |
| 11 | Soybean-mintill | 737 | 264 | 1454 | 2455 |
| 12 | Soybean-clean | 178 | 56 | 359 | 593 |
| 13 | Wheat | 62 | 26 | 117 | 205 |
| 14 | Woods | 380 | 136 | 749 | 1265 |
| 15 | Buildings-Grass-Trees-Drives | 116 | 34 | 236 | 386 |
| 16 | Stone-Steel-Towers | 28 | 5 | 60 | 93 |
| Total | | 3081 | 1025 | 6143 | 10,249 |

**Table 5.** The number of training, validation, test, and total samples in the UP dataset.

| No. | Class | Train | Val | Test | Total Samples |
|-----|-------|-------|-----|------|---------------|
| 1 | Asphalt | 1327 | 670 | 4634 | 6631 |
| 2 | Meadows | 3730 | 1810 | 13,109 | 18,649 |
| 3 | Gravel | 420 | 241 | 1438 | 2099 |
| 4 | Trees | 613 | 333 | 2118 | 3064 |
| 5 | Painted metal sheets | 269 | 134 | 942 | 1345 |
| 6 | Bare Soil | 1006 | 500 | 3523 | 5029 |
| 7 | Bitumen | 266 | 133 | 931 | 1330 |
| 8 | Self-Blocking Bricks | 737 | 363 | 2582 | 3682 |
| 9 | Shadows | 190 | 97 | 660 | 947 |
| Total | | 8558 | 4281 | 29,937 | 42,776 |

**Table 6.** The number of training, validation, test, and total samples in the KSC dataset.

| No. | Class | Train | Val | Test | Total Samples |
|-----|-------|-------|-----|------|---------------|
| 1 | Scrub | 153 | 78 | 530 | 761 |
| 2 | Willow swamp | 49 | 29 | 165 | 243 |
| 3 | CP hammock | 52 | 28 | 176 | 256 |
| 4 | Slash pine | 51 | 31 | 170 | 252 |
| 5 | Oak/Broadleaf | 33 | 18 | 110 | 161 |
| 6 | Hardwood | 46 | 22 | 161 | 229 |
| 7 | Swamp | 21 | 4 | 80 | 105 |
| 8 | Graminoid marsh | 87 | 45 | 299 | 431 |
| 9 | Spartina marsh | 104 | 39 | 377 | 520 |
| 10 | Cattail marsh | 81 | 40 | 283 | 404 |
| 11 | Salt marsh | 84 | 39 | 296 | 419 |
| 12 | Mud flats | 101 | 61 | 341 | 503 |
| 13 | Water | 186 | 87 | 654 | 927 |
| Total | | 1048 | 521 | 3642 | 5211 |

*4.2. Experimental Parameter Discussion*

We focused on two main factors that affect the classification performance of our proposed network, i.e., the ratio of the training dataset, and the number of groups $G$ of the residual group channel-wise attention (RGCA) module. Finally, according to the results of experiments, the ratios of the training, validation, and test datasets for the IN, UP, and KSC datasets are {3 : 1 : 6, 2 : 1 : 7, 2 : 1 : 7}, respectively. Additionally, we set the number of groups of RGCA for three datasets to 8. Furthermore, the spatial input size of the network was constantly set to 16 × 16 for all experiments.

4.2.1. Effect of Different Ratios of the Training, Validation, and Test Datasets

According to the experiment of the effect with different ratios of training samples in [28], we also divided the HSI datasets into four different rations (2 : 1 : 7, 3 : 1 : 6, 4 : 1 : 5, 5 : 1 : 4), and tested the impact of different numbers of training samples on our proposed model. To obtain accurate results with different training samples, we set the epochs of different ratios to {100, 100, 60, 60}, respectively. At the same time, the number of groups $G$ was 8. Finally, the training time, test time, and results of the three indicators (i.e., OA, AA, and kappa) with different ratios of the proposed model for three HSI datasets are list in Tables 7–9.

**Table 7.** Training time, test time, and OA under different ratios on the IN dataset by the proposed method.

| Ratios | Training Time (s) | Test Time (s) | OA (%) | AA (%) | Kappa × 100 |
|--------|-------------------|---------------|--------|--------|-------------|
| 2:1:7 | 10,861.78 | 99.90 | 99.52 | 99.22 | 99.53 |
| 3:1:6 | 15,769.93 | 85.99 | 99.87 | 99.88 | 99.85 |
| 4:1:5 | 12,320.78 | 72.52 | 99.86 | 99.77 | 99.84 |
| 5:1:4 | 15,138.30 | 59.03 | 99.86 | 99.74 | 99.82 |

**Table 8.** Training time, test time, and OA under different ratios on the UP dataset by the proposed method.

| Ratios | Training Time (s) | Test Time (s) | OA (%) | AA (%) | Kappa × 100 |
|--------|-------------------|---------------|--------|--------|-------------|
| 2:1:7 | 25,837.94 | 235.55 | 100.0 | 99.99 | 99.99 |
| 3:1:6 | 37,310.21 | 205.73 | 99.97 | 99.98 | 99.96 |
| 4:1:5 | 29,345.04 | 171.84 | 99.98 | 99.97 | 99.97 |
| 5:1:4 | 36,296.38 | 135.85 | 99.98 | 99.98 | 99.98 |

**Table 9.** Training time, test time, and OA under different ratios on the KSC dataset by the proposed method.

| Ratios | Training Time (s) | Test Time (s) | OA (%) | AA (%) | Kappa × 100 |
|--------|-------------------|---------------|--------|--------|-------------|
| 2:1:7 | 5142.95 | 47.34 | 100.0 | 100.0 | 100.0 |
| 3:1:6 | 7292.39 | 39.70 | 100.0 | 99.99 | 99.98 |
| 4:1:5 | 5779.27 | 33.82 | 99.98 | 99.98 | 99.99 |
| 5:1:4 | 7094.23 | 28.22 | 99.99 | 99.98 | 99.98 |

From Tables 7–9, we can find that in general, only with few training samples can our proposed network obtain high OA indicators in all three HSI datasets. Specifically, for the IN dataset, when the ratio changed from $2:1:7$ to $3:1:6$, the OA indicator showed a clear and large increasing trend from 99.52% to 99.87%, and in contrast, the training time increased less. When the number of training samples further increased, as the epochs of the training process were reduced from 100 to 60, the OA decreased slightly. Therefore, we chose the ratio of $3:1:6$ for the IN dataset. For the UP and KSC datasets, with the increasing number of training samples, the training time rose rapidly, whereas the accuracy decreased a little because of the epoch decreasing. Especially for the UP dataset, when the ratio was $2:1:7$, it had taken a long time to train the model. Additionally, when the ratio changed to $3:1:6$, the training time showed a dramatic jump from 25,837.94 s to 37,310.21 s, which nearly doubled. Additionally, with the further increase of the ratio, the corresponding training times were all longer than $2:1:7$. While for the KSC dataset, since it has fewer samples than the other two datasets, the training times of the different ratios were all lower, and when the ratio was $2:1:7$, the proposed model had already classified the KSC categories with a classification accuracy OA close to 100%. Therefore, for the UP and KSC datasets, we chose $2:1:7$ as the most suitable ratios. Furthermore, compared with the previous methods, our proposed network reached the highest classification accuracy; the detailed results will be shown later.

In addition, we may notice that the training time and training samples did not show a linear growth relationship. The reason is that the epochs of the four ratios were different, as mentioned above. Considering that when the ratio is $2:1:7$ or $3:1:6$, the number of the training samples is small, we therefore increased the epochs of $2:1:7$ and $3:1:6$ to 100 epochs to obtain the best results. While when the ratio came to $4:1:5$ or $5:1:4$, more training samples in each epoch could make the network achieve a high classification performance with fewer epochs.

### 4.2.2. Effect of the Number of Groups

In the proposed RGCSA network, in order to better optimize the channels, we divided the channels to $G$ groups and then used RCA module for each group, and finally merged the optimized channels of each group. Therefore, the number of groups is the other key factor for our proposed network. At the same time, since we utilized 3-D-ResNeXt to extract spectral-spatial features, for the convenience of the experiment, we set the cardinality (i.e., the size of the set of transformations) and $G$ to the same value. We evaluated the classification performance of the proposed RGCSA network with different numbers of groups $G$ and the results are shown in Table 10. In this experiment, we set the spatial input size to $16 \times 16$, and the ratio to $3:1:6$ for the IN dataset and $2:1:7$ for the UP and KSC datasets.

From the table, we can find that as the number of groups $G$ increases, the number of parameters and training time all gradually increase, while the OA indicators for the three HSI datasets fluctuate a little. It means that the reasonable division of the number channel groups is key to the influence on the classification performance of the network. If the number of groups $G$ is small, some channels may not be optimized, and even some useful channel information may be gradually discarded due to the deep network. If $G$ is too large, each group has fewer channels to be optimized, and the network cannot accurately extract useful channel information. Finally, we chose the number of groups $G = 8$ for three HSI datasets.

**Table 10.** Params, training time, test time, and OA for different numbers of groups *G* on the IN, UP, and KSC datasets.

| Datasets | *G* | Params | Training Time (s) | Test Time (s) | OA (%) |
|----------|-----|--------|-------------------|---------------|--------|
| IN | 6 | 2,974,912 | 11,923.85 | 64.96 | 99.54 |
|    | 8 | 4,489,120 | 15,769.93 | 85.99 | 99.87 |
|    | 10 | 6,264,960 | 21,401.03 | 113.47 | 99.87 |
| UP | 6 | 2,972,224 | 20,087.09 | 183.93 | 99.97 |
|    | 8 | 4,485,536 | 25,837.94 | 235.55 | 100.0 |
|    | 10 | 6,260,480 | 34,160.93 | 299.13 | 99.94 |
| KSC | 6 | 2,973,760 | 3887.32 | 36.16 | 99.97 |
|     | 8 | 4,487,584 | 5142.95 | 47.34 | 100.0 |
|     | 10 | 6,263,040 | 6734.43 | 58.56 | 99.97 |

*4.3. Classification Results Comparison with State-of-the-Art*

To verify the effectiveness of our proposed RGCSA network, we compared RGCSA with several classic methods, i.e., SVM [11], SSRN [21], 3-D-ResNeXt [28], DBMA [32], and SSAN [35]. To obtain fair comparison results, our proposed RGCSA network and compared methods adopted the same spatial input size of $16 \times 16 \times b$ (*b* represents the number of spectral bands), the ratio of 3 : 1 : 6 for the IN dataset, and 2 : 1 : 7 for the UP and KSC dataset for all methods.

Tables 11–13 report the OAs, AAs, kappa coefficients, and the classification accuracy for each class for three HSI datasets. From the tables, we can see that the proposed RGCSA achieved the highest classification accuracy than other methods for all three HSI datasets. First of all, SVM achieved the lowest classification accuracy in the three HSI datasets among all the methods. Secondly, since the training samples of class 1, 7, and 9 (alfalfa, grass-pasture-mowed, and oats, respectively) in the IN dataset are lower than 50 and SSRN divided the network into spectral and spatial feature learning parts, 2-D-CNN-based SSRN showed a lower classification accuracy on these classes, especially class 9, than other 3-D-CNN-based methods, such as 3-D-ResNeXt, DBMA, SSAN, and RGCSA. Though DBMA and SSAN used 3-D-CNN to extract features, these two models separated the spectral dimension and spatial dimension. It means that these models cannot make use of the close correlation between these two dimensions, and the results of DBMA and SSAN in Tables 11–13 proved it. Thirdly, we find that the models combined with the attention modules can achieve high classification accuracy, especially for the UP and KSC datasets. It means that channel-wise attention and spatial-wise attention can indeed optimize the features extracted by CNN and improve the classification performance. Furthermore, compared with these methods, our proposed RGCSA network could classify all classes for the three datasets more accurately, with classification accuracies higher than 99.80%. It means that our proposed network only needs fewer training samples to obtain higher classification performance through our proposed group-channel and space joint attention mechanism.

Figures 10–12 show the visualization maps of all classes of all methods based on CNN (i.e., SSRN, 3-D-ResNeXt, DBMA, SSAN, and our proposed RGCSA), along with the false color images of the HSI datasets and their corresponding ground-truth maps. In the IN dataset, the visualization maps of SSRN, DBMA, and SSAN did not show class 9 (oats, labeled in dark red). However, our proposed RGCSA network could classify class 9, which is completely displayed in Figure 10g. In the UP and KSC datasets, we find that the edge contours of each class of our proposed network are clearer and smoother than others. In addition, the prediction effect of our proposed RGCSA network on unlabeled parts is also significantly better than other methods. For example, on the right of class 1 (labeled by bright red) in the IN dataset, it can be seen from the false color map in Figure 10a that this unlabeled part should belong to class 6. None of the comparison methods can accurately predict this part. In contrast, it can be clearly seen from Figure 10g that our proposed RGCSA network can predict this part and fully visualize it. Similarly, in the lower middle of the UP dataset, the part marked in dark blue belongs to class 3 (gravel), and our proposed network can visualize it more completely than other

methods. In summary, our proposed RGCSA network can clearly visualize all the labeled classes, and can predict and visualize the unlabeled parts more accurately than other methods.

**Table 11.** Classification results of different methods for the IN dataset.

|  | SVM | SSRN | 3D-ResNeXt | DBMA | SSAN | RGCSA |
|---|---|---|---|---|---|---|
| OA (%) | 81.67 | 99.46 | **99.79** | 98.19 | 98.64 | **99.87** |
| AA (%) | 79.84 | 93.05 | **99.71** | 96.31 | 97.45 | **99.92** |
| Kappa × 100 | 78.76 | 99.39 | **99.70** | 97.94 | 97.50 | **99.85** |
| 1 | 96.78 | 100.0 | 100.0 | 100.0 | 100.0 | 100.0 |
| 2 | 78.74 | 100.0 | 100.0 | 97.10 | 97.65 | 100.0 |
| 3 | 82.26 | 99.00 | 99.80 | 99.03 | 98.69 | 99.80 |
| 4 | 99.03 | 98.59 | 98.59 | 92.20 | 96.95 | 99.29 |
| 5 | 93.75 | 99.65 | 99.30 | 99.26 | 99.15 | 100.0 |
| 6 | 85.96 | 100.0 | 100.0 | 98.20 | 98.95 | 100.0 |
| 7 | 40.00 | 100.0 | 100.0 | 81.25 | 97.65 | 100.0 |
| 8 | 91.80 | 100.0 | 100.0 | 100.0 | 100.0 | 100.0 |
| 9 | 0 | 0 | 100.0 | 85.71 | 89.94 | 100.0 |
| 10 | 96.00 | 97.44 | 100.0 | 98.00 | 99.14 | 100.0 |
| 11 | 70.94 | 99.73 | 99.59 | 98.46 | 99.12 | 99.79 |
| 12 | 74.73 | 99.72 | 99.72 | 98.15 | 98.95 | 99.87 |
| 13 | 99.04 | 100.0 | 100.0 | 100.0 | 100.0 | 100.0 |
| 14 | 94.29 | 99.74 | 100.0 | 99.74 | 99.96 | 100.0 |
| 15 | 85.11 | 100.0 | 100.0 | 96.12 | 98.14 | 100.0 |
| 16 | 96.78 | 95.00 | 98.31 | 97.67 | 97.33 | 100.0 |

**Table 12.** Classification results of different methods for the UP dataset.

|  | SVM | SSRN | 3D-ResNeXt | DBMA | SSAN | RGCSA |
|---|---|---|---|---|---|---|
| OA (%) | 90.58 | **99.97** | 99.93 | 98.88 | 99.05 | **100.0** |
| AA (%) | 92.99 | **99.96** | 99.91 | 98.71 | 98.91 | **99.99** |
| Kappa × 100 | 87.21 | **99.96** | 99.91 | 98.50 | 98.64 | **100.0** |
| 1 | 87.24 | 99.85 | 99.85 | 99.37 | 99.45 | 99.98 |
| 2 | 89.93 | 100.0 | 99.99 | 99.73 | 99.84 | 100.0 |
| 3 | 86.48 | 100.0 | 99.59 | 99.16 | 98.68 | 100.0 |
| 4 | 99.95 | 99.95 | 100.0 | 98.21 | 99.21 | 100.0 |
| 5 | 95.78 | 99.89 | 100.0 | 100.0 | 98.16 | 100.0 |
| 6 | 97.69 | 99.97 | 100.0 | 97.45 | 98.36 | 100.0 |
| 7 | 95.44 | 100.0 | 100.0 | 1000 | 99.11 | 100.0 |
| 8 | 84.40 | 100.0 | 99.77 | 95.12 | 98.26 | 100.0 |
| 9 | 100.0 | 100.0 | 100.0 | 99.36 | 99.12 | 100.0 |

**Table 13.** Classification results of different methods for the KSC dataset.

|  | SVM | SSRN | 3D-ResNeXt | DBMA | SSAN | RGCSA |
|---|---|---|---|---|---|---|
| OA (%) | 80.29 | **99.97** | 99.67 | 99.72 | 99.62 | **100.0** |
| AA (%) | 65.64 | **99.95** | 99.30 | 99.42 | 99.53 | **99.99** |
| Kappa × 100 | 77.98 | **99.97** | 99.63 | 99.50 | 99.58 | **99.99** |
| 1 | 92.16 | 100.0 | 100.0 | 100.0 | 100.0 | 100.0 |
| 2 | 86.16 | 99.40 | 99.38 | 97.16 | 98.41 | 100.0 |
| 3 | 42.55 | 100.0 | 97.40 | 98.45 | 97.65 | 100.0 |
| 4 | 67.69 | 100.0 | 99.40 | 100.0 | 99.45 | 99.99 |
| 5 | 0 | 100.0 | 95.76 | 1000 | 100.0 | 100.0 |
| 6 | 54.71 | 100.0 | 100.0 | 99.58 | 99.69 | 100.0 |
| 7 | 0 | 100.0 | 100.0 | 100.0 | 100.0 | 100.0 |
| 8 | 65.12 | 100.0 | 100.0 | 99.56 | 100.0 | 100.0 |
| 9 | 67.82 | 100.0 | 100.0 | 100.0 | 100.0 | 99.99 |
| 10 | 93.4 | 100.0 | 100.0 | 100.0 | 100.0 | 100.0 |
| 11 | 100.0 | 100.0 | 100.0 | 99.89 | 99.42 | 100.0 |
| 12 | 83.75 | 100.0 | 100.0 | 100.0 | 100.0 | 100.0 |
| 13 | 100.0 | 100.0 | 100.0 | 97.75 | 99.29 | 100.0 |

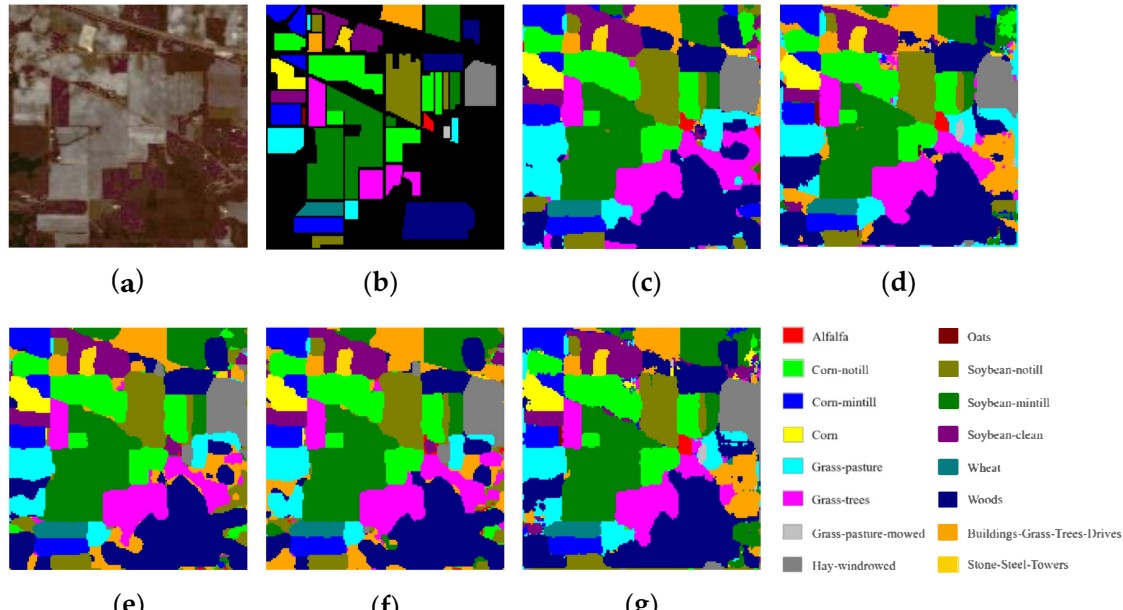

**Figure 10.** Classification results of the models in comparison with the IN dataset. (**a**) False color image. (**b**) Ground-truth labels, (**c–g**) Classification results of SSRN, 3-D-ResNeXt, DBMA, SSAN, and RGCSA.

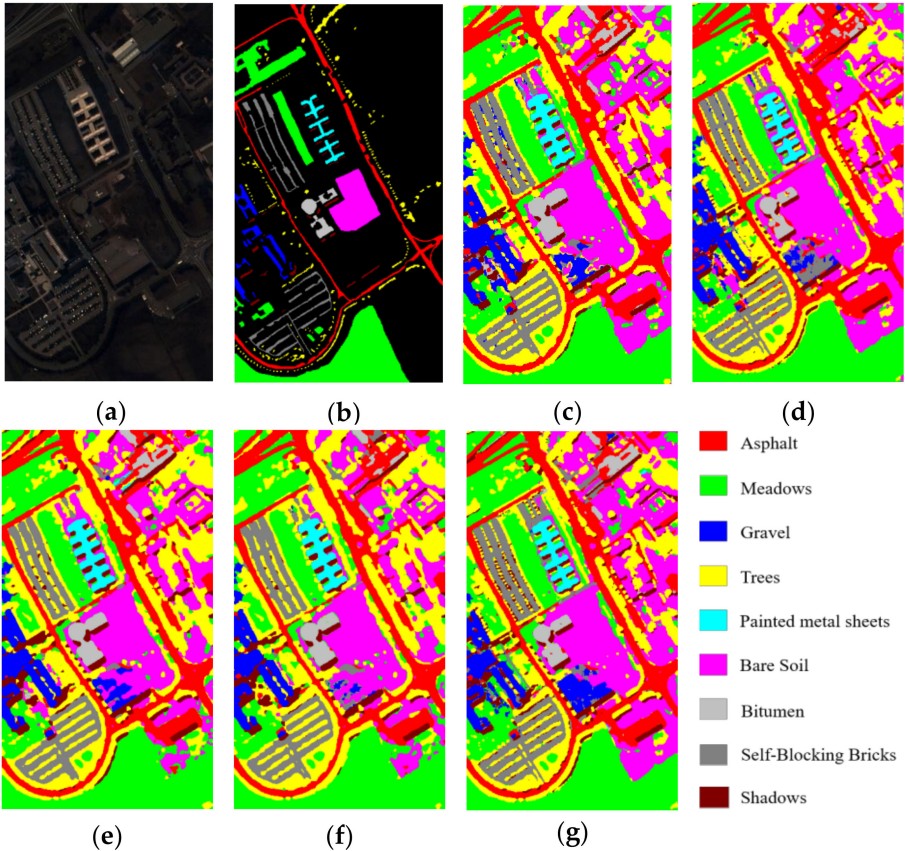

**Figure 11.** Classification results of the models in comparison with the UP dataset. (**a**) False color image. (**b**) Ground-truth labels, (**c**–**g**) Classification results of SSRN, 3D-ResNeXt, DBMA, SSAN, and RGCSA.

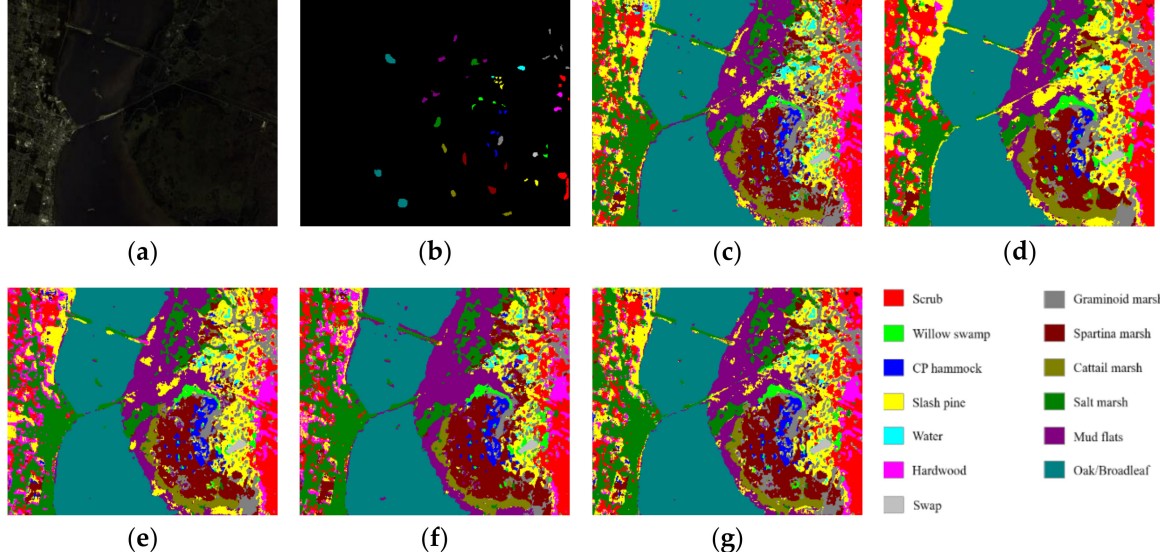

**Figure 12.** Classification results of the models in comparison with the KSC dataset. (**a**) False color image. (**b**) Ground-truth labels, (**c**–**g**) Classification results of SSRN, 3D-ResNeXt, DBMA, SSAN, and RGCSA.

Table 14 and Figure 13 show the comparison results of the 3D-ResNeXt (without the channel-wise and spatial-wise attention modules), RGCA (only with the group channel-wise attention RGCA module), RSA (only with the spatial-wise attention RSA module), and RGCSA (both with the RGCA module and RSA module). Except for the different attention modules, we set these four models to the same structure. The ratios of the three HSI datasets are {3 : 1 : 6, 2 : 1 : 7, 2 : 1 : 7}, respectively, and the

epochs were all set to 100. Table 14 shows the corresponding network parameters, training time, and test time and Figure 13 shows the OAs of four different attention mechanisms on the three HSI datasets. We find that the models with attention modules need more time to train, and the proposed RGCSA has the longest training time. Although RGCA, RSA, and RGCSA all generate more computational costs and consumption, from the figure, we can see that 3-D-ResNeXt without any attention modules achieved the lowest accuracies on the three HSI datasets, which proves the effectiveness of the attention mechanism. The OAs of RGCA with only the group channel-wise attention module in the three datasets are all higher than those of RSA with only the spatial-wise attention module, but the gap between RGCA and RSA is not obvious. It means that the proposed attention mechanisms in these two dimensions have optimized the channel and spatial features. Furthermore, when combining these two attention modules, the proposed RGCSA obtained the highest classification accuracies. It fully demonstrated that the proposed channel space joint attention mechanism plays an important role in HSI classification and is suitable for HSI classification tasks.

**Table 14.** Params, training time, and test time for different attention mechanisms on the IN, UP, and KSC datasets.

| Datasets | Methods | Params | Training Time (s) | Test Time (s) |
|---|---|---|---|---|
| IN | 3D-ResNeXt | 1,554,288 | 3054.29 | 20.68 |
| | RGCA | 2,736,992 | 10,629.34 | 55.43 |
| | RSA | 3,290,400 | 12,593.92 | 71.47 |
| | RGCSA | 4,489,120 | 15,769.93 | 85.99 |
| UP | 3D-ResNeXt | 1,550,704 | 6077.30 | 54.85 |
| | RGCA | 2,733,408 | 18,297.08 | 155.33 |
| | RSA | 3,286,816 | 18,801.38 | 184.43 |
| | RGCSA | 4,485,536 | 25,837.94 | 235.55 |
| KSC | 3D-ResNeXt | 1,552,752 | 1253.42 | 10.33 |
| | RGCA | 2,735,456 | 3584.96 | 30.23 |
| | RSA | 3,288,864 | 4044.99 | 38.97 |
| | RGCSA | 4,487,584 | 5142.95 | 47.34 |

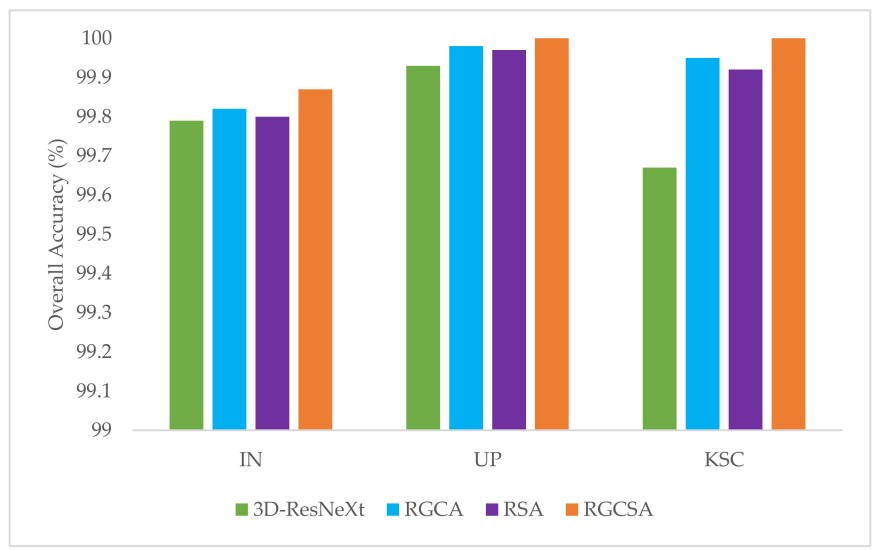

**Figure 13.** OAs of four different attention mechanisms on the three HSI datasets.

To test the robustness and generalizability of the proposed RGCSA under different ratios of training datasets, three models, i.e., SSRN based on 2-D-CNN, 3-D-ResNeXt based on 3-D-CNN, and the proposed RGCSA based on 3-D-CNN and attention mechanism, were selected to do this experiment. Figures 14–16 illustrate the overall accuracies (OAs) of these models using different ratios

of training datasets. When the number of training samples is small, such as the ratios of 2 : 1 : 7 and 3 : 1 : 6 in the three HSI datasets, the proposed RGCSA network obtained the highest OA indicators among the three methods. Especially for the IN and KSC datasets, the OAs of our proposed network always maintain a high level under different ratios. It means that we can achieve better classification results by the proposed RGCSA network with fewer training samples. It is important that when the total number of samples is small, or when there are few samples of some classes, such as class 1, 7, and 9 in the IN dataset, the proposed RGCSA can still generate a superior classification performance. As the training samples increase, the OAs of the proposed network in the three HSI datasets show a slight fluctuation but can still maintain over 99%. Since SSRN divided the network into the spectral learning part and spatial learning part, it cannot make full use of the relationship between the spectral and spatial dimensions. Therefore, in the three HSI datasets, the OAs of SSRN are the lowest among the three methods. It means that 3-D-CNN can extract spectral-spatial features with more useful information than 2-D-CNN. When compared with 3-D-ResNeXt, which needs more training samples to achieve high classification accuracies, our proposed network benefits from the residual attention mechanism to obtain the same high OA indicators with fewer training samples, and the classification accuracies are all higher than 3-D-ResNeXt under different ratios. In summary, it is obvious that the proposed channel-wise and spatial-wise attention modules, which can pay attention to the informative features, strengthen the representation of these features, and suppress the interference of useless information, are more suitable for HSI classification tasks. It can be demonstrated that our proposed RGCSA network has strong robustness and stability under different ratios of training datasets.

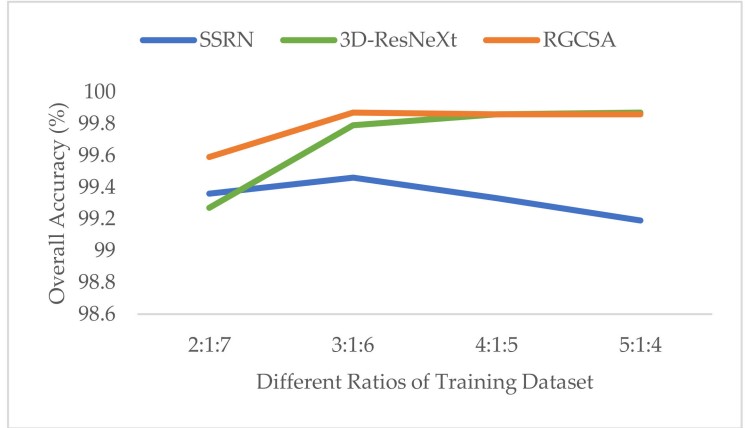

**Figure 14.** OAs of the SSRN, 3-D-ResNeXt, and RGCSA with different ratios of training samples for the IN dataset.

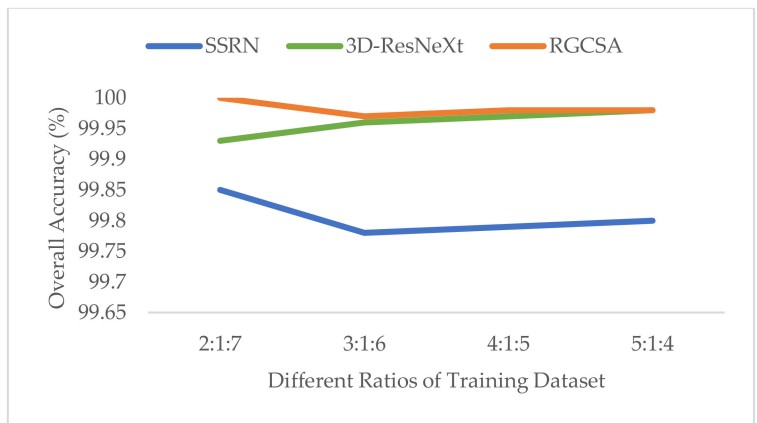

**Figure 15.** OAs of the SSRN, 3-D-ResNeXt, and RGCSA with different ratios of training samples for the UP dataset.

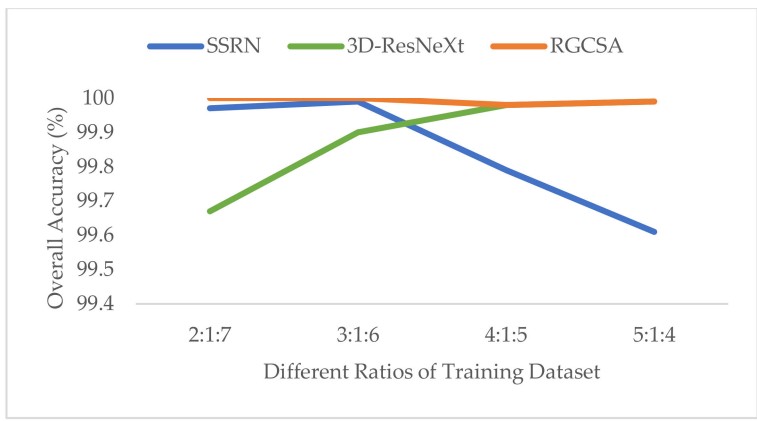

**Figure 16.** OAs of the SSRN, 3-D-ResNeXt, and RGCSA with different ratios of training samples for the KSC dataset.

At the same time, Tables 15–17 show the training time and test time of the above three models under different ratios. The epochs of the corresponding ratios are {100, 100, 60, 60}, respectively. From the tables, we can see that our proposed network needs more training time and test time in the three HSI datasets. The reason is that the proposed RGCA and RSA completely use 3-D-CNNs instead of FC layers to extract channel information and spatial context information. The proposed network is much deeper than the other networks, which results in more time to train the model. However, from the perspective of the classification results, it is feasible to exchange more computational costs for higher classification accuracies.

**Table 15.** Training time and test time for different networks in the IN dataset.

|  | Method | 2:1:7 | 3:1:6 | 4:1:5 | 5:1:4 |
|---|---|---|---|---|---|
| Training Time (s) | SSRN | 942.89 | 1059.21 | 1110.62 | 1262.90 |
|  | 3D-ResNeXt | 2966.77 | 3054.29 | 2974.87 | 4035.33 |
|  | RGCSA | 10,861.78 | 15,769.93 | 12,320.78 | 15,138.30 |
| Test Time (s) | SSRN | 9.63 | 8.36 | 7.25 | 5.51 |
|  | 3D-ResNeXt | 25.87 | 20.68 | 16.65 | 14.75 |
|  | RGCSA | 99.90 | 85.99 | 72.52 | 59.03 |

**Table 16.** Training time and test time for different networks in the UP dataset.

|  | Method | 2:1:7 | 3:1:6 | 4:1:5 | 5:1:4 |
|---|---|---|---|---|---|
| Training Time (s) | SSRN | 2469.97 | 2821.26 | 2750.48 | 3282.24 |
|  | 3D-ResNeXt | 6077.30 | 7095.93 | 6857.42 | 8410.72 |
|  | RGCSA | 25,837.94 | 37,310.21 | 29,345.04 | 36,296.38 |
| Test Time (s) | SSRN | 25.01 | 23.24 | 17.83 | 13.41 |
|  | 3D-ResNeXt | 54.85 | 52.18 | 39.32 | 36.74 |
|  | RGCSA | 235.55 | 205.73 | 171.84 | 135.85 |

**Table 17.** Training time and test time for different networks in the KSC dataset.

|  | Method | 2:1:7 | 3:1:6 | 4:1:5 | 5:1:4 |
|---|---|---|---|---|---|
| Training Time (s) | SSRN | 447.09 | 643.90 | 498.93 | 610.44 |
|  | 3D-ResNeXt | 1253.42 | 1384.68 | 1345.30 | 1632.73 |
|  | RGCSA | 5142.95 | 7292.39 | 5779.27 | 7094.23 |
| Test Time (s) | SSRN | 4.62 | 4.06 | 3.31 | 2.55 |
|  | 3D-ResNeXt | 10.33 | 8.66 | 7.22 | 5.75 |
|  | RGCSA | 47.34 | 39.70 | 33.82 | 28.22 |

## 5. Conclusions

In this paper, we proposed a supervised 3-D deep learning framework for HSI classification, using the bottom-up top-down attention structure with the residual connection. Compared with the previous traditional ML-based methods, the end-to-end deep learning methods can make full use of the GPU performance to accelerate network training and avoid complex artificial design of feature extraction. Compared with the deep learning methods based only on CNN, the proposed attention mechanism could strengthen important information and weaken unimportant information.

The above experiments verify the effectiveness of the proposed residual group channel and space attention module in the HSI classification tasks. In summary, the three major differences between our proposed RGCSA classification model and other deep learning-based models are as follows: First, the designed residual group channel-wise attention module and spatial-wise attention module have the same basic structure, which is easily inserted into any networks. Additionally, the residual connection can accelerate the flow of information for better training. Second, the group channel-wise attention module can reduce the possibility of losing useful information during the attention optimization. Additionally, the novel spatial-wise attention module can learn the context information and maintain the mapping relationship of spatial pixels before and after the optimization process. Third, the most important point is that in the face of poorly distributed HSI datasets, we can use the proposed RGCSA to optimize the learning process and obtain a higher classification performance with fewer training samples. In summary, the above advantages enable our RGCSA network to gain higher classification results than previous models, whether it is a CNN-based network or an attention-based network.

From the perspective of network complexity, the future work will focus on how to effectively reduce the number of network parameters and computational costs while maintaining high classification accuracy through the attention mechanism.

**Author Contributions:** P.W. and Z.C. conceived and designed the experiments; P.W. and Z.G. presented tools and carried out the data analysis; P.W. and Z.C. wrote the paper. Z.C. and F.L. guided and revised the paper. All authors have read and agreed to the published version of the manuscript.

**Funding:** This work was supported by National Natural Science Foundation of China (NSFC) (61501260) and "1311 Talent Program" of NJUPT.

**Acknowledgments:** 

**Conflicts of Interest:** The authors declare no conflict of interest.

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
