# Peer review of "Residual Group Channel and Space Attention Network for Hyperspectral Image Classification"

_remotesensing, doi:10.3390/rs12122035_

Round 1

Reviewer 1 Report

My feedbacks are summarized as follows:

  1. Apart from the technology overview presented in the first section, the authors also presented some related works in the second section. As some of the technologies have been widely applied in various fields today and Remote Sensing, I would like to suggest that the authors can perhaps compress some of the information or move it to the supplementary in order to focus on the applications.
  2. In the introduction, the authors have summarized three contributions of this paper. My suggestion is that the authors can perhaps specify the reasons and advantages of selecting HSI as the source of classification.
  3. Also, is the developed RGCSA applicable to other remote sensing data (ex. Photogrammetry, optical satellite and so on), as the authors mentioned in the Conclusion that this is a supervised 3-D deep-learning framework for HIS classification? If the this model is applicable to other remote sensing data, what are the advantages of using HSI?
  4. In the paper, the authors mentioned that the proposed HIS classification architecture is “end-to-end pixel-level” (Line 182 of page 4) and that the 16x16 patch is used as the input. I would like to know if the pixel level is determined based on the HSI dataset and what is the actual size of the 16 x 16 patch in the space.
  5. It is interesting to see the dataset samples listed by the authors in Tables 4-6. However, as they are considered as an important reference for the deep-learning training module, it will be important to know the future applications thereof.
  6. In Tables 7-9, the authors have specified the training time of this model. I understand that the training time is considered as an important criterion for a module’s efficiency, but it is even more interesting to know the accuracy of the proposed module. I would therefore suggested that the authors can give some feedback on dataset verification.
  7. English usage: The authors have definitely presented this article with excellent English writing skills. However, there are some minor suggestions that I would like to make:
  • Lines 16、17; 18、21; 159、160: The authors have repeatedly used the same adverb in nearby sentences. It is suggested that the authors can perhaps rephrase the sentences or use a different adverb.
  • Line 127、139、144、166、220、636、679: It is suggested that the authors can perhaps avoid using “And” to start a sentence.
  • Lines 139、199、211、280、564: English typo. Please check.
  • Lines 253-256: It is suggested that the authors can perhaps rephrase this sentence to better present their ideas.
  • Line 735: punctuation.

I have marked some of the sentences/words in this article: yellow represents only the important information from my point of view, whereas red are parts that are suggested for a revision.

Author Response

Please refer to the reply comments in the attachment.

Reviewer 2 Report

In this paper, the authors proposed a 3DCNN-based Residual Group Channel and Space Attention Network for Hyperspectral Image (HSI) classification.

Although, there have a lot of research work of HSI classification using deep learning based and other machine learning based approaches, the authors can clearly explain the main contributions and the differences of their approach with other approaches.

Moreover, the proposed network achieved the superior performance than the other state-of-the-art approaches on three publicly available HSI datasets.

Therefore, I strongly agree that this paper is suitable for publish in Remote Sensing.

I have one suggestion that the authors should provide the URL which can download the detailed parameters and network architecture of the proposed method. So that, the other researcher can replicate the experiments and compare the results in the future.

Author Response

(The authors gave the same response as above.)

Reviewer 3 Report

The paper is well written and the goal is clear. Despite this, it would be appropriate to improve it in some points:

The paper deals with a clustering and segmentation approach of pixels where the image is seen as a graph.

Should you cite the following paper that uses deep learning for graphs?

Ullah, I., Manzo, M., Shah, M., & Madden, M. (2019). Graph Convolutional Networks: analysis, improvements and results. arXiv preprint arXiv:1912.09592.

Related work should better capture the state of the art. It should be reorganized.

How is the accuracy of the experimental phase calculated? Are datasets supervised? Learn more about this very important aspect for the experimental part

Author Response

(The authors gave the same response as above.)
